# Aligning to Constraints for Data-Efficient Language Model Customization

## Abstract

General-purpose language models (LMs) are aligned to diverse user intents, but fall short when it comes to specific applications. While finetuning is the default method for customized alignment, human annotations are often unavailable in various customization scenarios. Based on the observation that one of the main issues of LM customization is constraint adherence, we investigate the feasibility of using constraints as a bridge from general LMs to customized ones. We investigate common constraints in NLP tasks, categorize them into three classes based on the types of their arguments, and propose a unified and efficient framework, ACT (Aligning to ConsTraints), for customizing LMs without human annotation. Specifically, ACT uses automatic constraint verifiers, which are typically easy to implement in practice, to compute constraint satisfaction rate (CSR) of each response. It samples multiple responses for each prompt and collects preference labels based on their CSR. Subsequently, ACT adapts the LM to the target task through a ranking-based learning process. Experiments on fine-grained entity typing, abstractive summarization, and temporal question answering demonstrate that ACT is capable of enhancing LMs' ability to adhere to different classes of constraints, thereby improving task performance comparable to or approaching that of finetuning with labeled data.

## 1 Introduction

General languages models (LMs) are aligned to diverse user instructions, but fall short when it comes to specific applications (Raffel et al., 2020; Ling et al., 2023; Saha et al., 2023). Customized alignment, which enables users to improve the task-specific capabilities of LMs, is therefore in high demand (Zhang et al., 2024; Lin et al., 2024; Zhou et al., 2024). To fullfil this goal, finetuning is the default method in LM services, such as GPT-4[1] and Gemini[2] finetuning APIs, which typically requires exhaustive human-annotated data. However, human annotations are often unavailable in various customization scenarios. Users have distinct purposes necessitating distinct annotations, but it is impractical to collect human annotations everytime due to budget limitation.

Recent research finds that the unsatisfactory adherence to task constraints is one of the main reasons for the failure of general LMs in downstream applications (Sun et al., 2023; Qin et al., 2024; Jiang et al., 2023; Abdin et al., 2023; Zhou et al., 2023b). Based on this observation, we investigate the feasibility of leveraging constraints to bridge the gap between general LMs and customized usages. Downstream applications typically contain explicit or implicit task constraints. For example, the fine-grained entity typing task has a label option list to define its decision space and a label hierarchy to describe the relation of sub-decisions (Fig. 1). These constraints contain informative task knowledge and can be automatically verified. On one hand, constraints produces informative supervision signals. They can help approximate the solution space, identify prediction errors, and guide the model toward the correct answer (Chang et al., 2007; Wang et al., 2023; Ning et al., 2018; Wang et al., 2020a). On the other hand, constraints enables efficient data collection. Assessing LM response quality with automatic constraint verifiers requires no human effort during annotation.

---

[1] https://platform.openai.com/docs/guides/fine-tuning
[2] https://ai.google.dev/docs/model_tuning_guidance

In this paper, we investigate common constraints in NLP tasks, categorize them into three classes based on the types of their arguments, and propose a unified and efficient LM customization framework, ACT (Aligning to ConsTraints), using automatic constraint verifiers to provide supervision signals for adapting models to downstream tasks (§3). As shown in Fig. 2, ACT starts from selecting constraints that can provide essential knowledge about user intents while at the same time automatically verifiable. Then, the constraint verifiers can efficiently measure constraint satisfaction rate (CSR) of model responses. These verifiers are typically easy to implement and are applicable to all instances governed by the corresponding constraints. With their assistance, ACT gathers supervision signals for LM adaptation based on unlabeled instances. It samples multiple responses for each unlabeled instance and automatically assigns relative preferences to them based on their CSR. Through a ranking-based learning process (Yuan et al., 2023; Liu et al., 2022), ACT integrates the knowledge revealed by the constraints into the LM.

We verify the effectiveness of our method on tasks with each class of constraints (§4), including fine-grained entity typing (Ling & Weld, 2012), abstractive text summarization (Narayan et al., 2018), and temporal question answering (Ning et al., 2020). Experimental results show that our method, even with little or no labeled data, can significantly enhance model capabilities on downstream tasks, achieving comparable performance to finetuning with the same amount of labeled data.

Our contributions are three-fold. First, we identify that downstream tasks often contain informative and auto-verifiable constraints. In this context, we formally define three classes of constraints that are beneficial to LM customization. Second, we propose ACT, a unified and efficient framework for customizing LMs, leveraging automatic constraint verifiers to produce supervision signals. Third, experimental results on various tasks and constraints demonstrate the effectiveness of our method across all classes of constraints.

Figure 1: An example of fine-grained entity typing with label option and label hierarchy constraints. A feasible response must satisfy both constraints.

## 2 RELATED WORK

**Constraints in NLP.** Constraints provide essential information about the detailed requirements of user intents, which widely exist in various NLP tasks, such as natural language inference (Roth & Yih, 2004; Minervini & Riedel, 2018; Li et al., 2019), information extraction (Ning et al., 2017; Wang et al., 2020a; Lin et al., 2023), and text summarization (Dou et al., 2021; Wang et al., 2022; Dixit et al., 2023). Constraints in these tasks range from simple fixed label options and format requirements to complex logic dependency (Faghihi et al., 2023). Prior works have integrated these constraints into artificial intelligent models through learning-based or inference-only methods, such as constraint driven learning (Chang et al., 2007; Minervini & Riedel, 2018), structured inference (Ning et al., 2017; Wang et al., 2023), and constrained decoding (Hokamp & Liu, 2017; Qin et al., 2022). Recent work also investigated integrating constraints into LMs to improve model performance on binary question answering (Burns et al., 2022; Jung et al., 2022) and natural language inference (Mitchell et al., 2022). Building upon these findings, we leverage automatic constraint verifiers for LM customization with an unified and efficient framework. Our framework makes no assumptions about the constraint type or source.

**LM Alignment and Customization.** LM alignment is crucial for LMs' capabilities in general scenarios (Zhang et al., 2023a; Ouyang et al., 2022; Mishra et al., 2022). However, aligning to general user instructions may not adequately improve LMs' capabilities in downstream use cases from unique and differentiated users. To enhance the task-specific capabilities of LMs, customization through finetuning is necessary (Zhang et al., 2024; Ling et al., 2023). Prior work on LM finetuning has explored various aspects, including parameter-efficient tuning (Dettmers et al., 2024), data curation

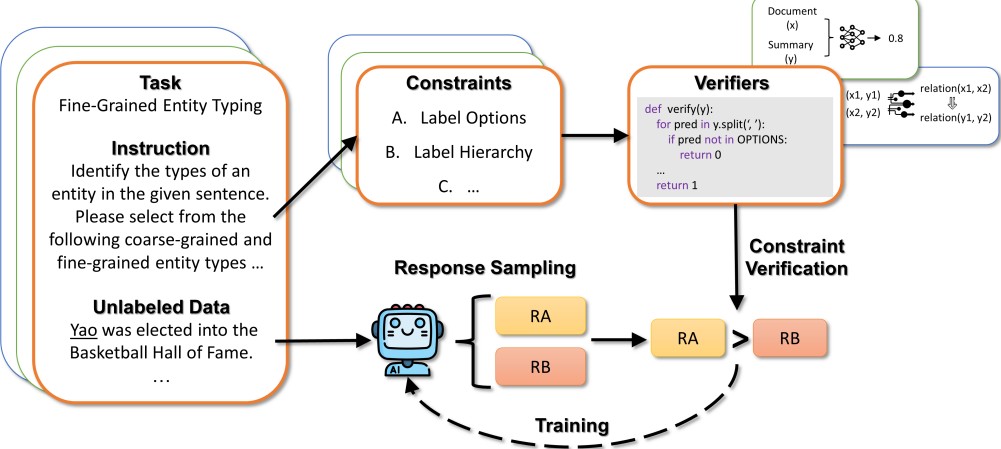

Figure 2: Overview of ACT. ACT utilizes automatic constraint verifiers, which are typically easy to implement in practice, to assess how well a response satisfies the constraints specified in the instruction. It samples two or more responses (e.g., RA and RB) for each prompt. Then, it computes the constraint satisfaction rate (CSR) of each response and assigns the preference label to each response pair based on their CSR (e.g., RA is better than RB). The preference labels serve as supervision signals for LM customization.

(Zhang et al., 2024), model selection (Lin et al., 2024), privacy protection (Yu et al., 2021), and safety issues (Qi et al., 2023). Some recent work has also finetuned task-specific reward models to adapt LMs (Wu et al., 2024). However, most of these works assume the availability of human-annotated data. When facing the data scarcity issue, there is no unified LM finetuning method that can be applied to various downstream tasks. Our work addresses the data issue in LM customization through the perspective of constraint satisfaction.

## 3 METHOD

We seek to build a unified framework to align LMs with various constraints. As shown in Fig. 2, the ACT framework starts from selecting proper constraints (§3.1) and implementing corresponding constraint verifiers (§3.2). Then, it samples multiple responses for each instance in the unlabeled task dataset (§3.3). The automatic constraint verifiers will measure the constraint satisfaction rate of responses and provide supervision signals for model alignment (§3.4). Finally, ACT aligns the model with constraints for adaptation (§3.5).

### 3.1 CONSTRAINT SELECTION

Formally, we define constraint as a function $f$ that verifies the satisfiablity of the prompt $x$ and the model response $y$. Derived from user instructions, they verify essential requirements for fulfilling user intents. According to the argument of $f$, we categorize task constraints into three classes:

- $f(y)$ defines a constraint for a response, such as response length, response format, and response candidate. For example, the fine-grained entity typing task requires the LM to respond with given options.
- $f(x, y)$ defines a constraint for a prompt-response pair. This type of constraint requires comparing the model input and output, such as their relevance and text overlap. For example, the abstractive summarization task expect a high relevance between the input document and the model-generated summary.
- $f(\{x_i, y_i\})$ defines a constraint for multiple prompt-response pairs. This type of constraint involves comparing multiple instances, such as the logical consistency of answers to related questions. For example, in temporal question answering, the answers to *"what happens before event A"* and *"what happens after event A"* should have no overlap.

In ACT, constraints should possess two properties: revealing *essential knowledge* and being *automatically verifiable*. Generally, constraints that more precisely approximate the user intent are more

effective in LM alignment. ACT can combine multiple constraints from different perspectives to achieve a more effective approximation.

## 3.2 VERIFIER REALIZATION

Constraint verifiers are the realization of $f$, measuring how well the response satisfies the constraints. They take the model response (and prompt) as the input, returning a constraint satisfaction rate (CSR). A higher CSR indicates that the response adheres to the constraints better. The verifiers can be rule-based (e.g., a function comparing words) or model-based (e.g., a relevance scorer), typically easy to implement from scratch or adapt from existing tools. In §4, we showcase the use of Python functions, model-based metrics, and rule engines as constraint verifiers. Note that each task may be associated with one or more constraints. Thus, the complete constraint verifier could be a combination of multiple sub-verifiers. The final CSR will be a weighted average of CSR from each sub-verifier, with the weights determined by the importance of the constraints.

## 3.3 RESPONSE SAMPLING

While a series of LM alignment studies have mentioned response sampling, little attention has been paid on improving the alignment effectiveness through decoding strageties. We draw inspiration from contrastive learning to gather high-quality negative responses (Robinson et al., 2021). The key to this step is ensuring that responses for the same unlabeled instance are distinguishable by the constraint verifiers (i.e., true negative), while simultaneously achieving high sampling probability (i.e., hard negative). If two responses have a close CSR, it could be challenging for even human annotators to decide which one is better. If the response with a low CSR also has a low sampling probability, penalizing it will not significantly benefit the model. In a nutshell, we seek to collect high-probability responses with non-negligible CSR gaps. Therefore, we employ decoding strategies that incorporate diversification and probability restriction, such as diverse beam search (Vijayakumar et al., 2018). This enables the collection of informative supervision signals in the next step.

## 3.4 CONSTRAINT VERIFICATION

Constraint verifiers can offer approximate but essential guidance for task adaptation, making them well-suited for the cost-efficient customization of LMs to specific tasks. ACT takes advantage of this property of automatic constraint verifiers to provide supervision signals for LM alignment. Specifically, the constraint verifier returns a CSR for each response or response combination. Then, we can assign preference labels to responses for the same prompt based on their CSR. For constraints defined over a single response or prompt-response pair, the response that has a higher CSR will be preferred. For example, in a task with label options constraint, a response within the option list is preferable to a response beyond it. For constraints defined over multiple prompt-response pairs, ACT creates a response combination by picking one response for each prompt. The constraint verifier computes the CSR for each response combination, and responses from the response combination with a higher CSR will be preferred. For example, when asking about events occurring before or after an event, the response combination that have no conflict (i.e., no overlap between the answers to 'before' and 'after') are preferable to those with conflicts. Then, each response will inherit the preference label of the combination it belongs to. As a result, ACT can collect preference labels from constraint verifiers as supervision signals to align models based on any type of constraints introduced in §3.1.

## 3.5 TRAINING

With the preference labels from constraint verifiers as supervision signals, ACT follows the learning objective of Yuan et al. (2023) with CSR as the reward. It encourages the model to generate the response with highest CSR for each prompt with

$$\mathcal{L}_{ft} = -\sum_i \log P(y_i | \mathbf{x}, \mathbf{y}_{<i}),$$

and optimizes a rank loss over all responses for the same prompt based on their relative CSR

$$\mathcal{L}_{rank} = \sum_{CSR_i < CSR_j} \max(0, P(\mathbf{y}^i | x) - P(\mathbf{y}^j | x)).$$

Since the CSR gap between each response pair may indicate fine-grained preference information, such as the relevance score in text summarization, we can further enhance the above loss functions. For $\mathcal{L}_{ft}$, we use CSR to reweight each datapoint. Because the quality of the best responses we sample for different prompts may vary, this strategy amplifies the impact of responses with higher CSR while reducing noise. For $\mathcal{L}_{rank}$, we use the CSR gap between each pair of responses as the ranking margin. This strategy allows the ranking loss to consider the relative preference, providing more informative supervision signals.

To further enhance learning efficiency, we adopt parameter-efficient tuning to align the LM with constraints. Specifically, we train LoRA modules (Hu et al., 2021) as customized adapters in a plug-and-play manner. The learning process is cost-efficient, and users have the flexibility to choose adapters based on constraints they need.

## 4 EXPERIMENT

In this section, we evaluate ACT on representative constraints for each of the three constraint categories introduced in §3.1, including fine-grained entity typing with label option and label hierarchy constraint ($f(y)$; §4.1), abstractive summarization with document-summary relevance constraint ($f(x, y)$; §4.2), and temporal question answering (QA) with the "no temporal conflict" constraint ($f(\{x_i, y_i\})$; §4.3).

### 4.1 $f(y)$: FINE-GRAINED ENTITY TYPING

**Task and Constraint.** Fine-grained entity typing seeks to select one or more applicable entity types of different granularities for an entity in a given sentence. We select two sub-constraints defined over the model response for this task: (1) label option, requiring all entity types to be selected from a fixed option list; and (2) label hierarchy, requiring to select a coarse-grained type if its corresponding fine-grained type is selected (e.g., an artist entity must be a person entity). Verifying these constraints needs to check the model output $y$. We implement the constraint verifier as a rule-based Python function, comparing the model response with the predefined label option and label hierarchy. Its pseudo code is in Appx. §A. In addition to entity typing, we further evaluate the lexical constraints in CommonGen (Lin et al., 2019) in Appx. §D, and compare ACT with constrained decoding, a representative inference-time intervention approach.

**Dataset and Metric.** We conduct experiments on the FIGER dataset (Ling & Weld, 2012) consisting of 112 entity types in two granularities. We sample 1K instances, which is the smallest effective data size used for LM alignment in prior studies (Jin et al., 2023; Zhou et al., 2023a), from the official training set as the unlabeled data, and five additional instances as in-context examples. For evaluation, we use the official test set. Following Ling & Weld (2012), we use macro-F1 over all instances as the evaluation metric. For this and the following tasks, we report the average result of three runs.

**Baselines.** We compare ACT with both training-free constraint integration and finetuning with labeled data. To integrate constraints into LMs, one way is *prompt w/ constraints* by adding verbalized constraints in the prompt. It adds into prompts the list of entity types with "`Label options: {all types}`" and the type dependency with "`If an entity is any of {fine-grained types}, it must also be {coarse-grained type}.`" The other way is *inference w/ constraints* through post-hoc correction.[3] The corrector is derived from the constraint verifier, correcting prediction errors according to the task constraints. *Finetuning* adopts the same instances used by ACT with human-annotated labels.

**Implementation Details.** For this and the following tasks, we use Falcon-7B-Instruct (Penedo et al., 2023) as the base model, because it is one of the few SOTA instruction-tuned LMs with Apache 2.0 license. We apply LoRA tuning in both ACT and finetuning. All models are trained using the same prompt templates and hyper-parameters in Appx. B and C. For each unlabeled instance, ACT collects multiple model responses through diverse beam search. Note that in this task, we consider a

---

[3]While other inference-time constraint integration approaches may also work, we do not observe significant difference in performance.

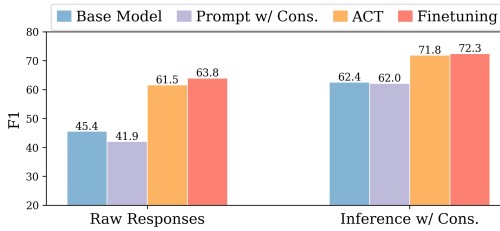 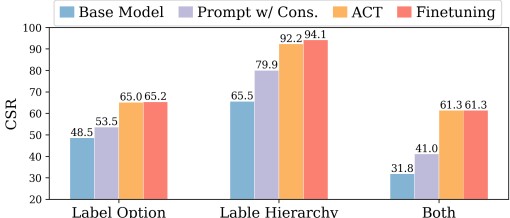

Figure 3: Results on fine-grained entity typing with $f(y)$ constraint. ACT, using supervision signals from automatic constraint verifiers, achieves performance close to that of *Finetuning* on the same amount of labeled data. *Inference w/ Constraints* is complementary to all the methods. Its improvement over ACT is much smaller, indicating constraints have been learned effectively.

Figure 4: Average CSR of raw responses on fine-grained entity typing. *Label Option* constraint limits the candidate set of entity types. *Label Hierarchy* constraint requires the answer to follow the hierarchy between coarse- and fine-grained entity types. A correct answer must satisfy *Both* constraints. ACT achieves CSR comparable to that of *Finetuning*.

binary CSR, selecting one response that satisfies all constraints and another that does not satisfies some constraints, for training. During the training and inference for all methods, we use the same five in-context examples.

**Results.** As shown in Fig. 3, ACT, with automatic feedback from constraint verifier, achieves comparable results to finetuning with human annotation on same amount of data. Further analysis in Fig. 4 shows that ACT achieves the same overall CSR as finetuning. These observations indicate that feedback from automatic constraint verifiers are effective surrogate of human feedback. Moreover, ACT can significantly improve the model's constraint-following capability with the help of automatic constraint verifiers. Although inference w/ constraints can further improve the performance of all methods as a complement, the improvement on ACT and finetuning are much smaller, indicating most of the knowledge about label constraints are already learned during training. Prompt w/ constraints improves model CSR, but does not improve the F1 score. We attribute this to the increased prompt length. Verbalizing the constraint adds several hundreds of tokens in the prompt, which unsurprisingly make it more difficult to understand.

## 4.2 $f(x, y)$: ABSTRACTIVE SUMMARIZATION

**Task and Constraint.** Abstractive summarization seeks to provide a brief summary for a given document. An essential constraint for this task is relevance – the information in the generated summary should be relevant to that in the given document. This constraint is necessary to achieve better factual consistency (Zhu et al., 2021; Dixit et al., 2023) and information coverage. To verify this constraint, we need to compare the model input $x$ and output $y$. We use BERTScore-Recall (Zhang et al., 2019) as the constraint verifier, because prior works have shown that it aligns well with the human judgement of summary quality and outperforms other metrics in downstream applications (Fabbri et al., 2021; Adlakha et al., 2023; Gupta et al., 2023). Note that we compute the BERTScore-Recall between the model response and the input document as CSR, which allows ACT to collect feedback with no human-annotated summary.

**Dataset and Metrics.** We conduct experiments on the XSUM dataset (Narayan et al., 2018), where each news article is paired with a human-written one-sentence summary. For training, we sample 1K instances from the official training set. We evaluate the model performance in a zero-shot manner. For automatic evaluation, we report ROUGE-L (Lin, 2004), BERTScore, and CSR. We further conduct **human evaluation** following the protocol in Zhang et al. (2023b). We recruit annotators from Amazon Mechanical Turk to label consistency (0 or 1), informativeness (5 point likert scale), and coherence (5 point likert scale) for system-generated and human-written summaries. Each summary is evaluated by three different annotators. The human evaluation instruction is in Appx. §G. Due to the computational and annotation cost, we sample 100 articles from the official test set for evaluation.

| Method | Training Data labeled : unlabeled | Automatic Evaluation | | Human Evaluation | | |
|---|---|---|---|---|---|---|
| | | BERTScore | ROUGE-L | Consistency | Informativeness | Coherence |
| Raw Model | - | 42.8 | 10.7 | 0.54 | 2.78 | 2.93 |
| Prompt w/ Cons. | - | 55.5 | 12.8 | 0.63 | 3.06 | 3.21 |
| Inference w/ Cons. | - | 58.9 | 13.6 | 0.56 | 2.87 | 3.07 |
| ACT | 0% : 100% | 65.1 | 15.7 | **0.68** | 3.12 | 3.35 |
| ACT | 10% : 90% | **68.6** | **18.2** | 0.65 | 3.20 | **3.44** |
| Finetuning | 100% : 0% | 68.2 | **18.2** | **0.68** | **3.24** | 3.40 |
| Ground-Truth | - | - | - | 0.81 | 3.66 | 3.81 |

Table 1: Automatic and human evaluation on abstractive summarization with constraint of $f(x, y)$ class. We also report the ratio of human-labeled and unlabeled training data for ACT and *Finetuning*. Note that *Inference w/ Constraints* is also applied to ACT and *Finetuning*, as they are complementary.

**Baselines.** *Prompt w/ constraints* emphasizes the relation between the summary and the input document in the prompt. *Inference w/ constraints* adopts the constraint verifier to rerank multiple sampled summaries, which is shown to outperform some training-based methods in prior work (Cao & Wang, 2021). *Finetuning* trains the LM with human-written summaries on the same training instances as ACT. Note that inference w/ constraints is complementary to other approaches, so we also apply it to ACT and finetuning.

**Implementation Details.** For ACT, we have two variants, with and without model warmup on 100 human-labeled data. With only a small amount of labeled data, the warm-up step enables the model to generate reasonable responses for a relatively complicated task, even though the model still achieves relatively low performance. We use the enhanced loss function, where $l_{ft}$ is re-weighted and $l_{rank}$ has a ranking margin. More details are in Appx. §C.

**Results.** As shown in Tab. 1, ACT with model warmup achieves comparable results in comparison with finetuning, and even outperforms the latter in terms of BERTScore in automatic evaluation and coherence in human evaluation. ACT with no human-labeled data, also performs as well as finetuning in terms of factual consistency. Both human and automatic evaluation indicate that aligning the model with the automatically verifiable relevance constraint can enhance the model performance on text summarization. Although model-generated summaries still have a

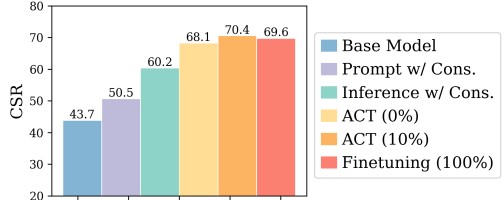

Figure 5: Average CSR of relevance constraint on model-generated summaries. ACT achieves even higher CSR than *Finetuning*.

gap with ground-truth summaries, it will not be difficult to scale up the size of training data for ACT with the help of the automatic constraint verifier. We further analyze model CSR in Fig. 5. ACT with warmup also outperforms finetuning from the perspective of constraint satisfaction. Both ACT and finetuning significantly outperforms the base model. This observation indicates a positive correlation between the quality of summaries and the adherence level to the summary-document relevance constraint.

### 4.3 $f(\{x_i, y_i\})$: TEMPORAL QA

**Task and Constraint.** Temporal question answering seeks to answer questions about the temporal relationship of events based on a given passage. Due to the nature of time, the responses to several interconnected questions should not have temporal conflicts. For example, the answers to *"what happens before event A"* and *"what happens after event A"* should have no overlap. Otherwise, an event may occur both before and after event A, leading to a time cycle. This constraint requires to compare multiple question-answer pairs $\{x_i, y_i\}$. We define a rule engine in Python as the constraint verifier, which identifies conflicts in temporal relationships among events.

---

[5]Since this experiment seeks to evaluate ACT on a specific class of constraints, we do not consider other stronger constraints. The "no temporal conflict" constraint only provides weak approximation of the answers. Thus, not supergisingly, further finetuning achieves better performance.

**Dataset and Metrics.** We conduct experiments on the TORQUE dataset (Ning et al., 2020), where each passage is paired with multiple temporal questions. We focus on the default set of questions which have clear logical relationships asking what happens before/during/after an event according to a given passage. We sample 1K group of questions from the official training set, leading to 3K instances in total. We report the average macro- and micro-F1 of three runs on the official development set.

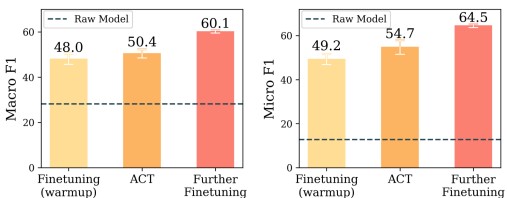

Figure 6: Results on temporal QA with constraint of $f(\{x_i, y_i\})$ class. As the raw model cannot generate reasonable answers, we use *Finetuning (warmup)* as the base model. ACT can even improve the performance of a finetuned model. *Further Finetuning* continually train the base model on labeled data.[5]

**Baselines.** Due to the complexity of the task and constraint, the raw model cannot generate reasonable responses and simply integrating constraints into the prompt or the inference process does not make the situation better. Therefore, we mainly compare our method with *finetuning* on human-annotated QA pairs.

**Implementation Details.** Since the base model fails to give reasonable answers, we apply model warmup for all methods. Specifically, we use 1K labeled data to warmup the model before ACT or further finetuning.[6] Then, ACT further tunes the model on 1K unlabeled data with feedback from the constraint verifier, while further finetuning adopts additional 1K human-labeled data. When collecting feedback from the constraint verifier, we sample 2 responses for each instance. Then, for all the $2^k$ response combinations of $k$ related questions, we use the constraint verifier to find one with no or the least conflicts as the preferred response combination. We use the preference label of the response combination as the preference label of each response within this combination. For all methods, we use the same three in-context examples. More details are in Appx. B and C.

**Results.** As shown in Fig. 6, the base model totally fails to give reasonable responses, revealing the difficulty of the task. ACT improves the performance of the warmuped model by 2.4 points in terms of macro-F1 and 5.5 points in terms of micro-F1. This indicates that ACT can even improve the performance of a finetuned model.

## 4.4 CONSTRAINT GENERALIZABILITY

To verify the generalizability of the learned constraint, we apply ACT to train and test the LM on different tasks with the same type of constraint. We conduct experiments on the extractiveness constraint, where the model response must be extracted from the input, and the relevance constraint introduced in §4.2. For the former, we evaluate constraint transfer among entity extraction, event trigger extraction, and slot extraction, while for the latter, we evaluate constraint transfer between text summarization and table-to-text generation. The results consistently show that the learned constraint knowledge is transferable across tasks.

**Extractiveness Constraint.** We select three tasks with this constraint: entity extraction, slot extraction, and event trigger extraction. The pseudo code of constraint verifier is in Appx. §A We use FIGER for entity extraction, MASSIVE (FitzGerald et al., 2022) for slot extraction, and ACE 2005 (Walker et al., 2006) for event trigger extraction. We sample 1K instances from each of MASSIVE and FIGER for training and 2K instances from ACE 2005 for evaluation. The CSR shows the model capability of following the target constraint. Prompts for all tasks adopt the same format with a constraint "You must extract the answer from the input sentence." During training and inference, we use five additional in-context examples. Detailed prompts and hyper-parameters can be found in Appx. B and C. Results in Fig. 7 show that the extractiveness constraint learned from entity extraction and slot extraction can be transferred to event trigger extraction, resulting in an improvement in CSR ranging from 8.9% to 17.4%, respectively. This indicates that the constraint-following capability is transferable. Combining multiple source tasks leads to better performance.

---

[6]The warmup step helps to mitigate the "garbage in, garbage out" problem, ensuring the availability of relatively good responses to facilitate informative feedback, particularly for complex tasks.

| Source Task | CSR on Target Task (T3) |
|---|---|
| - | 58.8 |
| Slot Extraction (T1) | 67.7 |
| Entity Extraction (T2) | 73.9 |
| Both (T1+T2) | 76.2 |

| Source Task | Target Task | R-L | BS |
|---|---|---|---|
| - | Summarization | 13.6 | 58.9 |
| Table-to-Text | Summarization | 15.6 | 62.3 |
| - | Table-to-Text | 21.1 | 60.0 |
| Summarization | Table-to-Text | 22.8 | 61.3 |

Figure 7: CSR of extractiveness constraint on event trigger extraction (T3). Learning the constraint from other tasks (T1 & T2) can improve the CSR on the target task.

Figure 8: ROUGE-L and BERTScore on summarization and table-to-text with the relevance constraint. Learning the constraint from one task can improve the performance on the other task.

**Relevance Constraint.** We further evaluate constraint transfer with the task (T1: text summarization) and constraint (relevance) in §4.2. We pair it with another task (T2: controlled table-to-text generation) with the same constraint. For T2, we use the ToTTo dataset (Parikh et al., 2020). The experiment setting is the same as §4.2. Results in Fig. 8 consistently show the transferability of learned constraints.

## 5 DISCUSSION

In this section, we delve into several topics about the generality of ACT and outline directions for future research.

### 5.1 CONSTRAINT ACCESSIBILITY

We have demonstrated in §2 that informative constraints are prevalent across various NLP tasks. Identifying constraints for a new task demands significantly less effort than manually annotating thousands of instances. The effort and expertise needed to define constraints and implement verifiers in ACT are comparable to those required for designing guidelines and setting up quality control pipelines for human annotation. In human annotation, annotators also must be aware of the task constraints, such as label options, beforehand. Without this knowledge, collecting high-quality data for learning purposes would be impossible. We posit that specifying constraints is a prerequisite for tasks requiring them, as humans must first understand the task constraints before annotation begins.

Constraints are prevalent in NLP tasks, and the extensive literature on these tasks serves as a valuable resource for identifying well-defined constraints (Roth & Yih, 2004; Minervini & Riedel, 2018; Li et al., 2019; Wang et al., 2020b; Parikh et al., 2020). At present, our approach relies on human efforts for constraint identification and verifier implementation. However, we envision the possibility of modularizing this process in the future. By combining different units, such as rule checkers and scorers, intelligent agents could potentially automate the creation of constraint verifiers, reducing the dependency on human intervention. This modular approach could streamline the workflow and expand the applicability of ACT to a broader range of tasks.

### 5.2 DISTRIBUTION OF CONSTRAINT SATISFACTION RATE

To understand the fine-grained behavior of ACT, we present the constraint satisfaction rate distribution for entity typing and summarization in Appx. §E, following the visualization style of Hong et al. (2024). The observation is that ACT and finetuning exhibit similar distributions, while the original model is significantly different.

### 5.3 CUSTOMIZING REWARD MODELS WITH ACT

While in this paper we focus on the standard finetuning process, which is the common practice of task adaptation for LMs, some recent studies have also adapted LMs with task-specific reward models (Wu et al., 2024; Stiennon et al., 2020). Our work does not use reward models as the main testbed because their training cost and stability hinder them from being widely adopted in LM services. The standard finetuning process effectively enables us to formulate the concept of ACT and prove its effectiveness on various tasks. Nonetheless, one can definitely customizing reward models with ACT. In Tab. 2, the experimental results show that ACT can also customize reward models achieving

| Reward Model | Prompt | Human Preference | | Constraint Preference | |
|---|---|---|---|---|---|
| | | Accurcy | Margin | Accurcy | Margin |
| No adaptation | w/o cons. | 26.3 | 0.3 | 17.3 | 0.5 |
| No adaptation | w/ cons. | 42.1 | 0.1 | 35.0 | 0.2 |
| ACT | w/o cons. | 82.0 | 7.0 | 86.4 | 4.9 |
| ACT | w/ cons. | 80.0 | 8.1 | 88.1 | 5.7 |
| Human annotation | w/o cons. | 87.5 | 7.2 | 79.4 | 5.0 |
| Human annotation | w/ cons. | 86.0 | 6.7 | 80.8 | 3.8 |

Table 2: Accuracy of response preference and average margin (between chose and rejected responses) of different reward models. We use ground-truth human preference and constraint-based preference as gold labels for evaluation. We evaluate reward models trained with human annotation and ACT. For each reward model, we have two variants, with and without verbalized constraints as input.

performance close to that of training with task-specific human preference. Although ACT is not originally proposed for adapting reward models, it can distill task constraint knowledge into reward models when human preference is unavailable.

We conduct experiments on fine-grained entity typing (§4.1) with a widely adopted reward model[7] in the huggingface hub. We use reward models with and without ACT to score and label the preference between human-annotated gold responses and model-generated incorrect responses. To show that ACT can achieve task adaptation performance close to methods with high-quality human annotation, we further train a task-specific reward model with task-specific human annotation for reference. We use two prompt variants, one with verbalized constraints (w/ cons.) and one without (w/o cons.). The results in Tab. 2 show that the general-purpose reward model fails on giving reliable scores for the downstream task, achieving an accuracy below 50%. It is also sensitive to the prompt, as adding verbalized constraints into the prompt can even lead to a 15.8 point performance drop. ACT increases the accuracy of preference labels to more than 80% with little human annotation. This result is close to training the reward model with task-specific human annotation.

To investigate reward models' ability of evaluating constraint satisfaction, we use them to score and label the preference between model responses satisfying and not satisfying constraints. ACT even outperforms the reward model finetuned with task-specific human annotation by up to 7.3 points. This highlights the effectiveness of ACT in incorporating prior knowledge of task constraints into models.

## 5.4    ACT AS A SERVICE

ACT presents a lightweight alternative to standard finetuning. With a predefined list of constraints, future LM services could offer APIs for LM customization based on ACT. In previous subsections, we have demonstrated that constraints are generally accessible and transferable. This enables service providers to store reusable constraints, constraint verifiers, and constraint-integrated adapters. Furthermore, future efforts can automate the selection of constraints and realization of verifiers. One potential approach involves retrieving constraints based on user instructions and then constructing verifiers by filling in templates.

## 6    CONCLUSION

In this paper, we propose an unified and efficient LM customization framework, ACT, aligning LMs to constraints for task adaptation. ACT leverages automatic constraint verifiers, which are typically easy to implement, to provide CSR as supervision signals. ACT can effectively enhance LMs' capability to adhere to task-specific constraints, thereby fulfilling the user intent for downstream application. We investigate common constraints in NLP tasks, categorize them into three classes based on the types of their arguments, and verify the effectiveness of ACT on all classes of constraints. Experiments on constraint transfer further shows the feasibility of tuning general constraint-following LMs. Future work may apply ACT to train compositional constraint adapters.

---

[7]OpenAssistant/reward-model-deberta-v3-large-v2

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

## A   CONSTRAINT VERIFIERS

We present the constraint verifiers in pseudo code of Python style.

**Label Option and Hierarchy.**

```python
# OPTIONS is a fixed list of valid options
# FINE2COARSE is a map from each
# fine-grained entity type to its
# corresponding coarse-grained entity type

def label_option(answers):
    for x in answers:
        if x not in OPTIONS:
            return 0
    return 1

def label_hierarchy(answers):
    for x in answers:
        if x not in FINE2COARSE:
            continue
        if FINE2COARSE[x] not in answers:
            return 0
    return 1

def constraint_verifier(response):
    answers = response.split(", ")
    first_cons = label_option(answers)
    second_cons = label_hierarchy(answers)
    return min(first_cons, second_cons)
```

**Extractiveness.**

```python
def constraint_verifier(inputx, response):
    csr = int(response in inputx)
    return csr
```

## B   PROMPT TEMPLATE

We follow the prompt template of Taori et al. (2023) for all experiments:

```
TEMPLATE

Below is an instruction that describes a task. Write a response that appropriately completes
the request.
### Instruction:
{$INSTRUCTION}

### Input:
{$INPUT}

### Response:
{$RESPONSE}
```

**Fine-Grained Entity Typing.**

---
**INSTRUCTION**

List all entity types of an entity in a given sentence.
Options: {$OPTIONS}.
If the entity is any of {$FINETYPES}, it is also {$COARSETYPE}.

---

---
**INPUT**

In the sentence {$SENTENCE}, what are the types of the entity {$ENTITY}?

---

**Abstractive Summarization.**

---
**INSTRUCTION**

Please generate a one-sentence summary for the given document.

---

---
**INPUT**

{$DOCUMENT}

---

**Temporal QA.**

---
**INSTRUCTION**

Select the best options to answer the question according to the passage.

---

---
**INPUT**

Passage: {$PASSAGE}
Question: {$QUESTION}
Options: {$OPTIONS}

---

**Constraint Transfer.**

---
**INSTRUCTION**

Identify the [entity / slot / event trigger] in the given sentence.
Your response must directly indicate the target information.
You must extract the answer from the input sentence.

---

---
**INPUT**

Which words indicate {$TYPE} in the sentence {$SENTENCE}.

---

## C HYPER-PARAMETERS

We use the same hyperparameters in all experiments unless otherwise specified.

**Training.** We train the models for 10 epochs with a batch size of 32 and a constant learning rate of 1e-5. We apply LoRA modules to the query, key, and value projectors in the attention module of each Transformer layer. The LoRA alpha, LoRA rank, and LoRA dropout are set to 16, 64, and 0.1 respectively. Following Yuan et al. (2023), we do not adjust the coefficient between $L_{ft}$ and $L_{rank}$, but simply add them. All inputs are left padded to 1,024 tokens. Note that we sampled 10% of the collected data for validation. For constraint transfer, we enlarge the size of LoRA modules and the learning rate to accommodate the shared constraint knowledge from different tasks. Specifically, we set LoRA alpha to 32, LoRA rank to 64, and constant learning rate to 2e-5.

**Inference.** During evaluation, we apply greedy decoding. For response sampling, we apply diverse beam search with four beams, four beam groups, and a diversity penalty of 1.

## D EXPERIMENTS ON COMMONGEN

We also compared constrained decoding and ACT on a subset of the CommonGen validation set. Constrained decoding achieved a ROUGH-L score of 41.6, while ACT, after less than 300 training steps, achieved a score of 42.0, further demonstrating the effectiveness of ACT. Additionally, we observed that the constraint satisfaction rate (CSR; i.e., concept coverage in this case) for constrained decoding is highly dependent on the beam size, whereas ACT can achieve a CSR of 92.3% without requiring further intervention. This highlights the different advantages of ACT and constrained decoding.

## E DISTRIBUTION OF CONSTRAINT SATISFACTION RATE

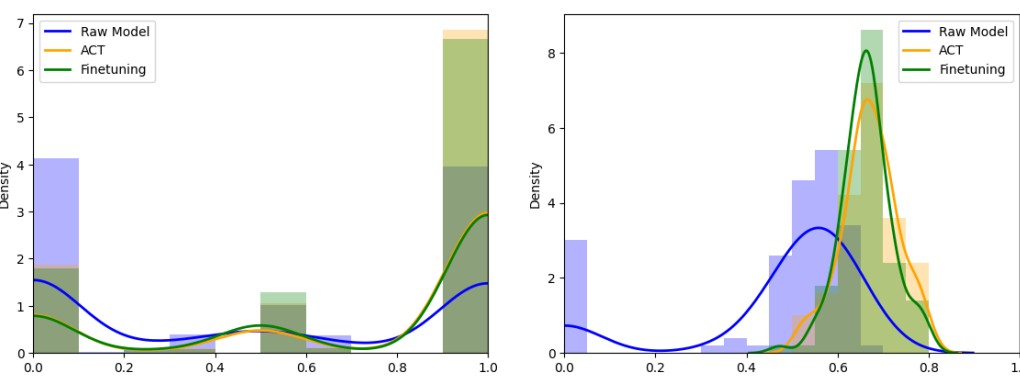

Figure 9: CSR distribution of entity typing.   Figure 10: CSR distribution of summarization.

## F LIMITATIONS

Due to license and accessibility restrictions, we cannot verify the effectiveness of ACT across a wide range of LMs. Despite the similarities in model structures and training processes among these LMs, variations in their implementation details may result in slightly different performance gains when applying ACT. Furthermore, while ACT notably reduces the cost of data collection for custom tasks, the steps involving constraint selection and verifier realization still require human effort. Automating these steps would contribute to further improvements. Finally, while our work demonstrates the potential of training various constraint-following adapters and general constraint-following models, we acknowledge that there is ample room for further exploration in this expansive area, providing opportunities for future research.

## G HUMAN EVALUATION

The interface including instructions for human evaluation is shown in Fig. 11.

**Instructions**

In this task, you will evaluate the quality of summaries written for a news article. Please carefully read the news article and the summaries. Then evaluate each summary from the following three perspectives.

- **Faithfulness**: Are the facts in the summary **consistent with** the facts in the news article? Please select between **"Yes" (faithful) and "No" (unfaithful)**.
- **Informativeness**: Does the summary capture the **key points** of the news article? Please rate on a scale from **1 (worst) to 5 (best)**.
- **Coherence**: Does the words in the summary follow a **coherent discourse**? Please rate on a scale from **1 (worst) to 5 (best)**.

**News Article:**

${Document}

**Summary A:** ${summaryA}

**Faithfulness:** Are the facts in the summary **consistent with** the facts in the news article? Please select between **"Yes" (faithful) and "No" (unfaithful)**.

○ **YES**  ○ **NO**

**Informativeness:** Does the summary capture the **key points** of the news article? Please rate on a scale from **1 (worst) to 5 (best)**.

○ **1**  ○ **2**  ○ **3**  ○ **4**  ○ **5**

**Coherence:** Does the words in the summary follow a **coherent discourse**? Please rate on a scale from **1 (worst) to 5 (best)**.

○ **1**  ○ **2**  ○ **3**  ○ **4**  ○ **5**

Figure 11: Human evaluation interface.

