# OpenReview forum: "Aligning to Constraints for Data-Efficient Language Model Customization"
_ICLR.cc/2025/Conference — ICLR 2025 Conference Withdrawn Submission_

### Official Review · Reviewer_dUjU · 2024-10-29

**Soundness:** 4
**Presentation:** 4
**Contribution:** 3
**Rating:** 6
**Confidence:** 3

**Summary:**

This paper is about incoporating constraints in LLMs in order to improve them efficiently for NLP downstream tasks (in the paper: summarization, entity typing and temporal question answering). The approach to do so relies - as far as I understand - a rather standard process in this area where constraints are defined (manually), automatically verified and different responses of an LLM are then generated, each of which is assessed for constraints. The model is then trained with these constraint-verified outputs. The paper goes on to show that this approach can rival standard fine-tuning of LLMs on three selected NLP tasks.

**Strengths:**

S1) The paper's idea is interesting

S2) The evaluation is thorough

S3) The paper is well-written and easy to follow

S4) The paper conducts multiple interesting and thorough analyses

**Weaknesses:**

I generally like this paper, but list here a few potential weaknesses:

W1) Even though I am not an expert for this, I think the methodology is straightforward and standard

W2) The topic is arguably not one of the most fanciest, concurrently

W3) Could one have added more baselines, e.g., constrained decoding?

**Questions:**

Q1) Choosing relevance as a constraint in summarization seems a bit arbitrary. Why not other aspects such as coherence? Overall, however, I am not sure whether it makes sense to think of those dimensions as constraint. Should that not be more formal aspects such as the length of the summary?

Q2) Did I miss something: In Table 3, you show the results for summarization&table-to-text generation but where is extraction? (Table 2 shows only CSR)

Q3) see W3

Depending on author answers, I am prepared to raise my score.

---

> ### Author Response · Authors · 2024-11-23
>
> We appreciate the reviewer's insightful feedback. We provide a detailed response below to address the concerns and questions raised by the reviewer.
>
> > **W1. the methodology is straightforward and standard**
>
> Our goal is **not to propose yet another alignment method but to introduce the insight** that alignment can be driven by constraints, specifically tailored for customized use cases. The research question we address is: If human annotation is unavailable and models cannot perform well on a task, where should the supervision signals come from for customized alignment? Our primary focus is to conceptualize and validate this across various scenarios. We are the **first to bridge LLM alignment with constraint-driven learning**. To support this, we formally categorize existing constraints into three types, demonstrate their integration into the alignment process, and showcase the effectiveness and transferability of representative constraints.
>
> > **W2. The topic is arguably not one of the most fanciest**
>
> Our work addresses a **critical and underexplored problem** in LM customization:
>
> Assume one has an off-the-shelf LM at hand, with no access to how this LM was trained by the provider (training data, RM model, etc.). However, this LM may not meet one’s needs in terms of constraint satisfaction, even if the constraint requirement is explicitly added to the prompts. The challenge is how to improve the LM's constraint adherence as cost-effectively as possible.
>
> - Approach 1: Post-processing — use the LM as-is and perform some post-editing to satisfy constraints (Inference with Constraints baseline in our paper).
> - Approach 2: Collect annotations for the task at hand and finetune the LM on this data (Finetuning as reference in our paper).
>
> This paper argues that both approaches are not ideal because Approach #1 underperforms, and Approach #2 is too costly. Thus, our approach focuses on the following problem: if the main issue with the LM is constraint-following, which can often be checked automatically, how do we automatically distill this knowledge into the LM?
>
> We believe our work offers a promising and impactful contribution to advancing the customized alignment of LLMs.
>
> > **W3. Could one have added more baselines, e.g., constrained decoding?**
>
> Constrained decoding is **orthogonal to our work** and is considered **one type of inference with constraints** in our paper. (In other words, constrained decoding is for inference, while ACT is for training.) We have already presented the results of such methods. We show that ACT and inference with constraints are complementary. By **combining our method with inference with constraints**, one **can achieve better performance**. As noted in Footnote 3, we have tested various inference-with-constraints methods and observed no significant performance difference. For consistency, we report the results of using inference with constraints derived from constraint verifiers.
>
> Per the reviewer's request, **we evaluated constrained decoding** on the entity typing task. The F1 score achieved is 64.0, which is slightly better than the inference w/ constraints result reported in the paper. Notably, when combining constrained decoding with ACT or finetuning, the F1 score improves significantly to over 72.0, demonstrating the effectiveness of our method.
>
> > **Q1. Choosing relevance as a constraint in summarization seems a bit arbitrary.**
>
> We chose relevance as a constraint because it is a key aspect of summary evaluation **identified in prior work** [1,2]. While coherence is also critical and length could be considered applicable, neither falls within the constraint category f(x, y) that this experiment aims to investigate.
>
> [1] Fabbri, Alexander R., et al. "Summeval: Re-evaluating summarization evaluation." TACL 2021.
>
> [2] Zhang, Tianyi, et al. "Benchmarking large language models for news summarization." TACL 2024.
>
> > **Q2. where is the results for extraction?**
>
> We emphasize the Constraint Satisfaction Rate (CSR) in our analysis to demonstrate the transferability of constraint adherence capabilities. Event trigger extraction has traditionally been framed as an NLU task rather than NLG, and there is no universally established metric for this specific context. However, in response to the user’s request, we report the exact match (EM) accuracy below.
>
> | Source Task | CSR   | EM   |
> |-------------|-------|------|
> | -           | 58.8  | 16.8 |
> | T1          | 67.7  | 18.2 |
> | T2          | 73.9  | 19.7 |
> | T1+T2       | 76.2  | 20.0 |

---

> ### Author Response · Authors · 2024-11-25
> **Followup**
>
> Dear Reviewer dUjU,
>
> Thank you for your time and thoughtful feedback on our paper. We hope our responses have addressed your concerns and kindly request you to consider raising your score. If there are any remaining issues, please let us know, and we will be happy to provide further responses.
>
> Thanks!

---

> > ### Comment · Reviewer_dUjU · 2024-11-26
> >
> > Dear authors,
> >
> > thanks for the answers - maybe making shorter ones could be a recommendation (reviewers have a lot of papers to review, so brevity can be important).
> >
> > I was/am generally optimistic for this paper, and as you see, I am the only reviewer that recommends acceptance right now. Mostly for this reason, and because I am not an expert for the topic, I am not increasing my score right now.
> >
> > But I keep leaning positive and your answers were also ok.
> >
> > > While coherence is also critical and length could be considered applicable, neither falls within the constraint category f(x, y) that this experiment aims to investigate.
> >
> > True. But this sounds like answering "we didn't do it" to a question "why didn't you do it"? My question is more like "why this constraints and not another one". There's no answer to the second part, or did I miss it?

---

> ### Author Response · Authors · 2024-11-28
>
> > **"why this constraints and not another one"**
>
> Thank you for raising this insightful question.
>
> **TL;DR:**
> *  It is not about selecting a specific constraint for a particular task but rather identifying a representative constraint for a broad category and then choosing a suitable task as the testbed for evaluation.
> * The coherent constraint falls into the f(y) type. For this type, we chose to verify a more general constraint—option lists, which are widely used to define valid solution spaces across various tasks.
>
>
> **Explanation**
>
> As indicated in lines 238-242, we select representative tasks for each of the three constraint categories to ensure comprehensive coverage of distinct types of constraints. It is not about selecting a specific constraint for a particular task but rather identifying a representative constraint for a broad category and then choosing a suitable task as the testbed for evaluation. Additionally, in Section 5.2, we conduct experiments to demonstrate the transferability of constraints across tasks, showcasing their adaptability and broader applicability. Overall, our work includes five groups of experiments that span a diverse spectrum of constraints, further validating the generality and robustness of our approach.
>
> The coherent constraint falls into the f(y) type. For this type, we chose to verify a more general constraint—option lists, which are widely used to define valid solution spaces across various tasks. This approach allows us to evaluate constraints that are broadly applicable and representative of real-world scenarios.

---

> > ### Author Response · Authors · 2024-12-04
> >
> > Dear Reviewer dUjU,
> >
> > As the discussion period is ending, we would like to thank you for volunteering your time and engaging in discussion. We appreciate your positive review of our paper and hope we have answered all your questions and addressed any concerns you had.
> >
> > Thanks!

---

### Official Review · Reviewer_HmdB · 2024-10-31

**Soundness:** 3
**Presentation:** 2
**Contribution:** 2
**Rating:** 5
**Confidence:** 4

**Summary:**

In this paper, the authors proposed a methodology to customize an LLM to specific user needs by introducing task constraints.
The main contributions consist of, firstly, defining a categorization of constraints that covers scenarios where the constraints involve the response alone (as in restricting the label(s) of the prediction), context and response, and groups of prompts and responses. Secondly,
the framework allows for the implementation of automatic constraint verifiers, which are used to build preference data and customize the models using a ranking-based method.
Finally, experiments on tasks representative of each constraint scenario, including analysis on transfer task learning scenarios, and the potential impact of constraint verification as a reward signal for reward modeling.

**Strengths:**

-	The framework makes no assumptions about the constrain type, and can be easily categorized into the proposed hierarchy.
-	The collection strategy for the preference dataset is sound, maximizing the “constraint satisfaction score” gap between responses whilst making sure all of them are highly probable. This effectively leverages the sensitivity of the “constraint satisfaction score” given for each scenario and the well-trained knowledge in the base LLM.

**Weaknesses:**

-	Although the constraint hierarchy is sound, only one task was investigated per constraint category. The exploration of at least two more tasks per category would provide a more confident indicative that the proposed method is efficient and we are not just overfitting to a specific task. Perhaps peripheral experiments in section 5 can be reworked as main experiments in section 4.
- Critically, the paper is also missing more details and examples on the defined constraint categorization, as well as an analysis on distribution of the constraint satisfaction scores for each constraint scenario (see ORPO, https://arxiv.org/pdf/2403.07691)

**Questions:**

## Section 3.5: Training
-	Some details are missing (as well as numeration on the equations) from the loss functions, which make the section difficult to understand for someone who hasn’t seen the cited paper (RRHF). It is alright to add redundant details in order to make the sections more understandable. For instance, the margin (L228) does not appear in L_rank.

## Section 4.1. Implementation details
-	From what I gather, the chosen (preferred) response is defined as one that satisfies all constraints, whereas a rejected response is one that does not satisfy ‘some’ constraints.  which kind of constraints are considered when selecting a rejected response?

## Section 4.2.
-	Details of the ‘enhanced loss function’ are missing in appendix C
-	For the ‘inference with constraints’ baseline, how often are the constraints verified during inference? After each token or sentence, or at the end?

## Section 4.3
-	Footnote 6, could you please elaborate in a more technical way what the ‘garbage in-garbage out’ problem is?
-	When collecting feedback from the constraint verifier, two responses are sampled from all possible combinations. It is desirable for this response pair to have minimal conflict. How is this conflict defined, and how is it quantified?

## Section 5
- 5.1. The argumentation in this section would be better placed at the related work section, as it provides general motivation
- 5.2. In table 2 and 3, when source task = ‘-‘, does this mean that training was done on the target task or that inference was done over the base model (before ACT training)

**Details Of Ethics Concerns:**

No concerns

---

> ### Author Response · Authors · 2024-11-23
>
> We appreciate the reviewer's insightful feedback. We provide a detailed response below to address the concerns and questions raised by the reviewer.
>
> > **W1. show that we are not just overfitting to a specific task**
>
> Our goal is to fit user-specified constraints, not tasks. The **generalizability of learned constraints has been shown in Section 5**. We agree with the reviewer that reframing the additional experiments in Section 5 as main experiments would better demonstrate the method's generalizability, and we will update our paper accordingly. Currently, we have five groups of experiments showing the constraint satisfaction rate for specific tasks and also the transferability of constraints, covering a diverse spectrum of constraints.
>
> > **W2.1. examples on the defined constraint categorization**
>
> For f(y) constraints, an example is shown in Figure 1, where the response must be selected from an option list (Section 4.1).
> For f(x,y) constraints, examples include the relevance constraint for summarization (Section 4.2) and the extractiveness constraint for information extraction (Section 5.2).
> For f({x,y}) constraints, an example is the consistency constraint for temporal QA (Section 4.3), where the answer to "happens before event A" and "what happens after event A" should have no overlap (lines 377-379).
> We will add concrete examples in the appendix in the updated paper shortly.
>
> > **W2.2. analysis on distribution of the constraint satisfaction scores**
>
> Per the reviewer's request, we present the **constraint satisfaction rate distribution for [entity typing](https://bashify.io/i/nedJO9) and [summarization](https://bashify.io/i/hj30Iy)**. (Please check the links pointing to anonymous figures.) The observation is that ACT and finetuning exhibit similar distributions, while the original model is significantly different.
>
> We will add this analysis in the updated paper and acknowledge ORPO for inspiring this insightful analysis.
>
> > **Section 4.1: Which kind of constraints are considered when selecting a rejected response?**
>
> If the response does not satisfy one or more of the given constraints, it will be rejected.
>
> > **Section 4.2: Enhanced loss function**
>
> $$ \mathcal{L}_{ft} = - CSR \sum_i \log P(y_i|\mathbf{x}, \mathbf{y}_{<i})$$
>
> $$\mathcal{L}_{rank} = \sum_{CSR_i < CSR_j} \max (0, P(\mathbf{y}^i|x) - P(\mathbf{y}^j|x) + CSR_i - CSR_j)$$
>
> > **Section 4.2: How often are the constraints verified during inference?**
>
> The constraints are verified at the end, following Cao & Wang (2021).
>
> Cao and Wang. "CLIFF: Contrastive learning for improving faithfulness and factuality in abstractive summarization." EMNLP (2021).
>
> > **Section 4.3: Footnote 6 – Could you please elaborate in a more technical way on what the ‘garbage in-garbage out’ problem is?**
>
> The original model cannot generate reasonable responses without any finetuning. In this case, no informative supervision signals can be collected by comparing the responses, as none of them are of high quality.
>
> > **Section 4.3: How is this conflict defined, and how is it quantified?**
>
> The conflict is defined as the overlap of events in two responses. It is quantified as the ratio of overlapping events.
>
> > **Section 5.2: When the source task = ‘-’, what does this mean?**
>
> This indicates that inference was done over the base model (before ACT training).

---

> > ### Author Response · Authors · 2024-11-26
> >
> > Dear Reviewer HmdB,
> >
> > Thank you for your time and thoughtful feedback on our paper. We hope our responses have addressed your concerns, and we kindly request that you consider updating the score accordingly. If there are any remaining issues, please let us know, and we will be happy to provide further clarification.
> >
> > Thanks!

---

> ### Author Response · Authors · 2024-12-04
>
> Dear Reviewer HmdB,
>
> As the discussion period is ending, we would like to thank you for volunteering your time to review our paper. We hope to have answered all your questions and addressed the rest of the concerns you had.
>
> Thanks!

---

### Official Review · Reviewer_mNHe · 2024-11-04

**Soundness:** 3
**Presentation:** 3
**Contribution:** 2
**Rating:** 5
**Confidence:** 3

**Summary:**

The paper presents an approach to adapt LLM to focus on a particular task which comes with constraints. The task is assumed to have a set of constraints, each of which are easy to automatically verify (e.g., you can write a simple code that checks whether the model output satisfies the constraint or not). The paper propose a simple, reasonable approach to address this issue: they build automatic constraint verifiers manually, sample multiple responses from the base LMs, evaluates the responses with constant verifiers, and fine-tune LLM to prefer constant satisfying responses over non constant satisfying ones.

 And I find the paper is well-written and easy to follow. The experiments are well explained, and I do not have major concerns with the validity of the proposed method. But I'm having an issue with the baselines and the gains from the proposed approach. Please see weaknesses for the details.

**Strengths:**

* They propose a simple and reasonable approach that can be applied to a wide range of applications.
* The task is well motivated.
* The experimental design is solid, though it misses baselines from prior work.
* The paper is clearly written.
* The discussion (Section 5) is quite thorough and interesting.

**Weaknesses:**

* Weak results(?):
 The gains from the proposed approach is pretty small or non-existent compared to reasonable baseline (e.g., fine-tuning approaches). For example, in fine-grained entity typing, fine-tuning outperforms (Figure 3&4). I'm not sure I'm understanding the Fine-tuning setup -- is ACT "unsupervised" (w/t access to human labels) and fine-tuning approach "supervised" with labeled data? Instaed, ACT has manually crafted constraint verifiers?  More elaboration on how much supervision is given to each setting would be helpful.

* Issues with baselines / task choice: It would be helpful if authors can provide justification for the choice of three benchmarks. There are other benchmarks that has been studied for constrained generations, for example, COMMONGEN corpus [1] which considers lexically constrained decoding (LLMs goal is to generate a coherent sentence containing all listed words). There are rich literature (which the paper cites e.g., COLD decoding paper) that handles such constraint adherence problems in this dataset, and they should be compared as a baseline.

[1] B. Y. Lin, W. Zhou, M. Shen, P. Zhou, C. Bhagavatula, Y. Choi, and X. Ren. CommonGen: A constrained
text generation challenge for generative commonsense reasoning. In EMNLP - Findings, 2020. URL https://aclanthology.org/
2020.findings-emnlp.165.pdf.

**Questions:**

(1) In your experiments, how many constraints (on average) exist for each task?
(2) There exist tasks where either (a) building a reliable verifier is difficult (e.g., long-form question answering) or (2) reliable verifier is computationally very expensive. Do you have any ideas how your approach could be modified/further improved to address such tasks?



Comments:
* I am not sure identification of three downstream tasks where constraints apply is strong enough to be listed as a contribution in the introduction.
* The verifier must be built manually, if I’m understanding correctly. It’d be good to make that clear in Figure 2.
* Human evaluation result (for summarization) should be more carefully reported, with inter-annotator agreement and statistics on the annotators, payment for them, etc.
* It would be helpful to have example task instance for each one in the appendix.

Minor comments/suggestions:
* For reproducibility, it’d be good to provide the exact prompt / in-context examples used (e.g., line 294).

---

> ### Author Response · Authors · 2024-11-23
>
> We appreciate the reviewer's insightful feedback. We provide a detailed response below to address the concerns and questions raised by the reviewer.
>
> > **W1. The gains are small compared to fine-tuning**
>
> We want to clarify that **fine-tuning is not a baseline** but rather an **“upper-bound” reference**. While ACT does not require human-annotated data, fine-tuning relies on human annotations (of the same data size), as mentioned in lines 280-282. The comparison with fine-tuning demonstrates the informativeness of feedback from constraint verifiers and the effectiveness of constraint-driven alignment.
>
> ACT only requires pre-defined constraint verifiers to automatically generate supervision signals. As discussed in Section 5.1, **identifying constraints demands significantly less effort than manual annotation** and is similar to designing annotation guidelines.
>
> > **W2.1. It would be helpful if authors can provide justification for the choice of three benchmarks.**
>
> As indicated in lines 238-242, we select representative tasks for **each of the three constraint categories**. We also conduct additional experiments to demonstrate constraint transferability in section 5.2. Overall, we present five groups of experiments, covering a diverse spectrum of constraints.
>
> > **W2.2. constrained decoding should be compared as a baseline**
>
> Constrained decoding is **orthogonal to our work** and is considered **one type of inference with constraints** in our paper. (In other words, constrained decoding is for inference, while ACT is for training.) We have already presented the results of such methods. We show that ACT and inference with constraints are complementary. By **combining our method with inference with constraints**, one **can achieve better performance**. As noted in Footnote 3, we have tested various inference w/ constraints methods and observed no significant performance difference. For consistency, we report the results of using inference w/ constraints methods derived from constraint verifiers.
>
> Per the reviewer's request, **we evaluated constrained decoding** on the entity typing task. The F1 score achieved is 64.0, which is slightly better than the inference-with-constraints result reported in the paper. Notably, when combining constrained decoding with ACT or finetuning, the F1 score improves significantly to over 72.0, demonstrating the effectiveness of our method.
>
> We will update our paper soon to include a discussion of [1] and COLD.
>
> > **Q1. how many constraints exist for each task**
>
> To avoid confounding factors, we focus on at most two constraints of the same type for each experiment and report the constraint satisfaction rate for each independent constraint.
>
> > **Q2. Do you have any ideas how your approach could be further improved to address such tasks ( where verifiers are difficult to build or are expensive to run)?**
>
> As discussed in Section 5.1, implementing constraint verifiers demands **significantly less effort** than manual data annotation and is similar to designing annotation guidelines. Constraint verifiers are **reusable** (even across tasks, as shown in Section 5.2) and can generate substantial supervision signals after their initial implementation. Additionally, given a new user request, one can **retrieve** the corresponding constraint verifiers or the trained constraint adapters for efficient alignment, as discussed in section 5.4.
>
> We thank the reviewer for the insightful comments and suggestions, which can help us improve the quality and clarity of our paper.

---

> > ### Comment · Reviewer_mNHe · 2024-11-27
> >
> > Hi, thank you for many clarifications. This helps!
> >
> > Regarding the constraint decoding baselines.
> > Thank you for the clarification! Adding this discussion would strengthen the paper. However, I still want to see comparison with these orthogonal, well-established approaches. You do not necessarily have to outperform them all -- especially as your approach can be combined with existing approaches. However, you should contextualize your approach with these existing literature: what is the pros/cons of inference time intervention vs. fine-tuning approaches like yours? In this regard, you should provide results not only on entity-typing but on other benchmarks as well, and evaluate your approach in the commongen benchmark (or other benchmarks that other papers have evaluated their approaches on).
> >
> > I am not sure I understand the answers to the my second question. I am not sure you can easily claim "implementing constraint verifiers demand significantly less efforts" than data annotation efforts. For complex tasks (e.g., long form question answering, or creative text generation), I have hard time envisioning constructing a constraint verifier. Could you elaborate on this point? I think this is a limitation of this approach, and should be discussed appropriately.

---

> ### Author Response · Authors · 2024-11-26
>
> Dear Reviewer mNHe,
>
> Thank you for your time and thoughtful feedback on our paper. We hope our responses have addressed your concerns, and we kindly request that you consider updating the score accordingly. If there are any remaining issues, please let us know, and we will be happy to provide further clarification.
>
> Thanks!

---

> ### Author Response · Authors · 2024-11-28
>
> > **Regarding the constraint decoding baselines.**
>
> We appreciate the reviewer’s insightful feedback. The key advantage of ACT-style tuning methods over inference-time interventions lies in their ability to enhance the model’s capabilities, whereas inference-time interventions rely on utilizing the existing model capabilities. This partially explains why the two methods are complementary.
>
> In our paper, we evaluate different inference-time intervention methods based on the literature related to each task. For entity typing, we assessed post-hoc rule-based correction and constrained decoding. For summarization, we employed reranking beams at the last decoding step, a proven method that is effective in boosting model performance (Cao and Wang, 2021). The results align: ACT can further boost model performance. Notably, ACT can improve constraint satisfaction rates, achieving performance levels comparable to inference-time interventions.
>
> We also compared constrained decoding and ACT on a subset of the CommonGen validation set. Constrained decoding achieved a ROUGH-L score of 41.6, while ACT, after less than 300 training steps, achieved a score of 42.0, further demonstrating the effectiveness of ACT. Additionally, we observed that the constraint satisfaction rate (CSR; i.e., concept coverage in this case) for constrained decoding is highly dependent on the beam size, whereas ACT can achieve a CSR of 92.3% without requiring further intervention. This highlights the different advantages of ACT and constrained decoding.
>
>
> Cao and Wang. "CLIFF: Contrastive learning for improving faithfulness and factuality in abstractive summarization." EMNLP (2021).
>
> > **implementing constraint verifier can be a limitation**
>
> Thank you for highlighting this concern. We apologize for any confusion caused. We agree that constructing constraint verifiers is a limitation of ACT, and we have discussed this limitation and proposed potential solutions in Appendix D.
>
> Constraints are prevalent in NLP tasks, and the extensive literature on these tasks serves as a valuable resource for identifying well-defined constraints [1-6] (more examples are available in lines 90-102). For creative text generation, for instance, a lexical-level creativity scorer [7] could be a potential tool for building constraint verifiers. Drawing an analogy to data annotation, we posit that specifying constraints is a prerequisite for tasks requiring them, as humans must first understand the task constraints before annotation begins.
>
> At present, our approach relies on human efforts for constraint identification and verifier implementation. However, we envision the possibility of modularizing this process in the future. By combining different units, such as rule checkers and scorers, intelligent agents could potentially automate the creation of constraint verifiers, reducing the dependency on human intervention. This modular approach could streamline the workflow and expand the applicability of ACT to a broader range of tasks.
>
> [1] Chang, Ming-Wei, Lev Ratinov, and Dan Roth. "Guiding semi-supervision with constraint-driven learning." ACL 2007.
>
> [2] Wang, Haoyu, et al. "Joint constrained learning for event-event relation extraction." EMNLP (2020).
>
> [3] Jang, Myeongjun Erik, and Thomas Lukasiewicz. "Consistency analysis of chatgpt." arXiv preprint arXiv:2303.06273 (2023).
>
> [4] Pan, Wenbo, et al. "A preliminary evaluation of chatgpt for zero-shot dialogue understanding." arXiv preprint arXiv:2304.04256 (2023).
>
> [5] Parikh, Ankur P., et al. "ToTTo: A controlled table-to-text generation dataset." EMNLP (2020).
>
> [6] Porteous, Julie, and Marc Cavazza. "Controlling narrative generation with planning trajectories: the role of constraints." ICIDS 2009.
>
> [7] Kuznetsova, Polina, Jianfu Chen, and Yejin Choi. "Understanding and quantifying creativity in lexical composition." Proceedings of the 2013 conference on empirical methods in natural language processing. 2013.

---

> > ### Comment · Reviewer_mNHe · 2024-12-02
> >
> > Thank you for the response.
> >
> > For the second task -- is there more recent baseline you can adapt than this one from 2021?
> >
> > Regarding the experiments on Commongen:
> > ---
> > We also compared constrained decoding and ACT on a subset of the CommonGen validation set. Constrained decoding achieved a ROUGH-L score of 41.6, while ACT, after less than 300 training steps, achieved a score of 42.0, further demonstrating the effectiveness of ACT. Additionally, we observed that the constraint satisfaction rate (CSR; i.e., concept coverage in this case) for constrained decoding is highly dependent on the beam size, whereas ACT can achieve a CSR of 92.3% without requiring further intervention. This highlights the different advantages of ACT and constrained decoding.
> > ---
> > How did you select a subset here? Why a subset? What constrained decoding method are you using here? These comparison to prior work is not thorough thorough enough. Can you compare with the numbers in prior studies?
> >
> > ====
> >  The key advantage of ACT-style tuning methods over inference-time interventions lies in their ability to enhance the model’s capabilities, whereas inference-time interventions rely on utilizing the existing model capabilities.
> > >> I am not sure I buy this argument. It depends on what you see as a "model". I think architecture + parameters + decoding method (which can be optimized) can be considered as a system/model. And it seems that sometimes inference time intervention (entity typing) can actually work better..

---

> > > ### Author Response · Authors · 2024-12-02
> > >
> > > We thank the reviewer for the followup.
> > >
> > > > **is there more recent inference-time baseline you can adapt than this one for summarization constraints?**
> > >
> > > As far as we know, this is the best-performing inference-time intervention for summarization. The limited improvement in recent work also highlights the difficulty of integrating constraints into generation tasks like summarization during inference, even with constraint verifiers. In contrast, our method does not rely on strong assumptions about constraints and is broadly generalizable and applicable.
> > >
> > > > **thorough comparison with prior work on CommonGen**
> > >
> > > We want to clarify that this additional experiment is intended purely as a proof of concept. To ensure efficiency, we randomly sampled 200 instances. We followed the constrained decoding approach outlined in [1]. Since the settings, such as the base model, differ significantly, there are no directly comparable numbers from prior work.
> > >
> > > [1] Post, Matt, and David Vilar. "Fast lexically constrained decoding with dynamic beam allocation for neural machine translation." arXiv preprint arXiv:1804.06609 (2018).
> > >
> > > > **I think architecture + parameters + decoding method (which can be optimized) can be considered as a system/model. And it seems that sometimes inference time intervention (entity typing) can actually work better.**
> > >
> > > Our claim is based on the general observation that, even with the same inference-time intervention, the best performance a model can achieve is highly correlated with its original performance. This happens not only in our case, but also in broad scenarios such as instruction following [1], ICL [2], and RAG [3]. This correlation explains why researchers are actively developing more advanced base models and post-training methods. One key message in the entity typing experiment is that ACT and inference-time interventions are complementary; in other words, ACT enhances the upper bound of what inference-time interventions can achieve.
> > >
> > > [1] Wei, Jason, et al. "Finetuned language models are zero-shot learners." arXiv preprint arXiv:2109.01652 (2021).
> > >
> > > [2] Brown, Tom B. "Language models are few-shot learners." arXiv preprint arXiv:2005.14165 (2020).
> > >
> > > [3] Soudani, Heydar, Evangelos Kanoulas, and Faegheh Hasibi. "Fine Tuning vs. Retrieval Augmented Generation for Less Popular Knowledge." arXiv preprint arXiv:2403.01432 (2024).

---

> ### Author Response · Authors · 2024-12-04
>
> Dear Reviewer mNHe,
>
> As the discussion period is ending, we would like to thank you for volunteering your time and engaging in discussion. We found your comments to be the most challenging and rewarding, motivating some significant changes and hopefully improvements to our paper. We hope to have answered all your questions and addressed the rest of the concerns you had.
>
> Thanks!

---

### Official Review · Reviewer_Fejv · 2024-11-07

**Soundness:** 2
**Presentation:** 3
**Contribution:** 2
**Rating:** 5
**Confidence:** 4

**Summary:**

# Goal an Method
This work looks at general problem of training LLMs to perform tasks where the output must fulfill some set of constraints. Their proposed training approach assumes access to some implemented constraint verification method for measuring how closely the LLM's output follows the task's constraints. Training is then done in a manner that is similar to rejection sampling: For a given, unlabeled input they first sample multiple outputs from the model before scoring and ranking outputs using their constraint verification method. The authors then update the LLM by training on two losses using these rankings: (1) the standard NLL loss of generating the best ranked sample and (2) ranking loss encouraging the model to assign greater likelihood to samples rank higher in constraint following than those that rank lower.

# Main Results
The authors experiment on a variety of tasks with different constraints: (1) Fine-Grained Entity Typing w/ Label Space and Label Hierarchy Constraints (2) Abstractive Summarization w/ Extractiveness Constraints and (3) TemporalQA w/ Event Ordering Logic Constraints.
The primary baselines used in this work are using the base LLM (w/ prompting for constraint following) and (2) finetuning on gold labels. The results demonstrate that their proposed method improves performance over simple prompting. They also demonstrate that in the first two settings, their proposed method can achieve comparable performance while reducing annotation cost. In the third setting, there is a still a significant gap between full finetuning and their proposed method.

# Additional Results
The authors demonstrate that the learned constraints can transfer across tasks (learning extractiveness constraint on table-to-text summarization transfers to text-to-text). The authors also experiment with using their constraint-following rankings to train reward models, and find it that it improves reward modeling performance on Fine-Grained Entity Typing, but still performs worse than training with gold, human-annotated labels.

**Strengths:**

The work is well written, clear, and technically sound. Results are generally positive, demonstrating that

**Weaknesses:**

# Generalizability and Reliance on Constraint Verifier
A significant limitation of the method proposed in this work is the reliance on the constraint verifier. This limits the utility of the proposed method, where constraint verifiers are applicable and implemented. This work also, for the most part, explores settings where constraint following is directly tied to the end task performance metric, which is often not the case. The only exception to this is the summarization setting, where the authors do find that some mixing between constraint following and standard finetuning performs best. [1] Presents an interesting additional setting and methods where constraints are not directly tied to end-task performance (human preference), and are input dependent.

[1] Rule Based Rewards for Language Model Safety
Tong Mu, Alec Helyar, Johannes Heidecke, Joshua Achiam, Andrea Vallone, Ian Kivlichan, Molly Lin, Alex Beutel, John Schulman, Lilian Weng

# Missing Comparisons Against Existing Methods
The proposed training approach is very similar to standard training methods like rejection sampling and preference learning methods (e.g., DPO) , where the constraint verification evaluator is used in lieu of a reward model. The most significant differences are seem to be minor changes to the learning objective (Loss function in L218 and L222), and response sampling method (Section 3.3). Describing the differences between the proposed training approach and experiments comparing the proposed method against these standard approaches and ablations using different response sampling methods would help determine the differences and the impact of these changes in moving from a reward-maximizing to constraint-following setting.

**Questions:**

Could you clarify the necessity of only evaluating on LLMs following the Apache 2.0 license and whether there are any other public LLMs that could be used for additional experiments? The limited model settings would be listed as a weakness of this work, but this is understandable if there is no further options.

How is Inference w/ constraints implemented? Is it the sampling approach as is used during training for generating candidates? I'm particularly curious of in the improvement of ACT (100% setting ) and inference w/ constraints performance in Table 1. It's also surprising that inference w/ constraints significantly improves entity typing performance, over ACT, but not in summarization. Is there any explanation for this?

---

> ### Author Response · Authors · 2024-11-23
>
> We appreciate the reviewer's insightful feedback. We provide a detailed response below to address the concerns and questions raised by the reviewer.
>
> >  **W1.1. generalizability and reliance on the constraint verifier**
>
> As discussed in Section 5.1, implementing constraint verifiers demands **significantly less effort** than manual data annotation and is similar to designing annotation guidelines. Constraint verifiers are **reusable** (even across tasks, as shown in Section 5.2) and can generate substantial supervision signals after the initial implementation.
>
> All constraints discussed in our paper are derived from general task guidelines, such as option lists, extractiveness, and consistency. **Constraints are widely observed in NLP tasks [2-7]** (more examples are available in lines 90-102), as discussed in Section 5.1. It is very unlikely that one cannot specify a constraint for a task unless the solution space is arbitrary text.
>
>
>
> > **W1.2. constraint following is directly tied to the end task performance metric, which is often not the case**
>
> We want to clarify that, in terms of utility, **any user-specified constraint is part of the task goal, narrowing down the solution space**, regardless of whether it is directly or indirectly related to the performance metric. A response that does not meet the constraints is unhelpful to the user. From this perspective, **[1]** (such as apology and refusal rules) also **falls into one type of constraint, f(x, y), as defined in our paper**.
>
> We want to kindly point out that [1] was released after the initial release of our paper. We will update our paper soon and add a discussion of this work. Note that [1] and our work have different focuses. We investigate common constraints in NLP tasks, categorize them into three classes, and propose a unified and efficient constraint-driven alignment framework.
>
> [1] Rule Based Rewards for Language Model Safety Tong Mu, Alec Helyar, Johannes Heidecke, Joshua Achiam, Andrea Vallone, Ian Kivlichan, Molly Lin, Alex Beutel, John Schulman, Lilian Weng
>
> [2] Chang, Ming-Wei, Lev Ratinov, and Dan Roth. "Guiding semi-supervision with constraint-driven learning." ACL 2007.
>
> [3] Wang, Haoyu, et al. "Joint constrained learning for event-event relation extraction." EMNLP (2020).
>
> [4] Jang, Myeongjun Erik, and Thomas Lukasiewicz. "Consistency analysis of chatgpt." arXiv preprint arXiv:2303.06273 (2023).
>
> [5] Pan, Wenbo, et al. "A preliminary evaluation of chatgpt for zero-shot dialogue understanding." arXiv preprint arXiv:2304.04256 (2023).
>
> [6] Parikh, Ankur P., et al. "ToTTo: A controlled table-to-text generation dataset." EMNLP (2020).
>
> [7] Porteous, Julie, and Marc Cavazza. "Controlling narrative generation with planning trajectories: the role of constraints." ICIDS 2009.
>
> > **W2. The proposed training approach is very similar to standard training methods**
>
> We want to clarify that our **research direction is orthogonal** to the mentioned training methods. The research question we address is: If human annotation is unavailable and models cannot perform well on a task, **where should the supervision signals come from** for customized alignment? Our **insight is that the alignment process can be constraint-driven**. We are the first to connect LLM alignment with constraint-driven learning. To support this insight, we formally categorize existing constraints into three types, demonstrate how to incorporate them into the alignment process, and show the effectiveness and transferability of representative constraints. Our findings can definitely be extended, and our constraint verifier can be integrated into other alignment processes.
>
> > **Q1. Could you clarify the necessity of only evaluating on LLMs following the Apache 2.0 license … this is understandable if there is no further options.**
>
> We make this technical choice due to institution-wide policy restrictions.
>
> > **Q2. How is Inference w/ constraints implemented?**
>
> As introduced in lines 265-277 and 344-346, inference with constraints is derived from constraint verifiers, but instead of assessing constraint satisfaction rate, we use them to improve the response. For summarization, we adopt the constraint verifier to rerank multiple sampled summaries, following Cao & Wang (2021).
>
> Cao and Wang. "CLIFF: Contrastive learning for improving faithfulness and factuality in abstractive summarization." EMNLP (2021).
>
> > **Q3. I'm particularly curious about the improvement**
>
> The improvement depends on the initial task performance and the properties of the constraints. Note that for entity typing and summarization, we use constraints from two different categories to demonstrate the generalizability of our framework. Our goal is not to achieve state-of-the-art performance but to formulate the concept of aligning with constraints and prove its effectiveness across a wide range of scenarios.

---

> ### Author Response · Authors · 2024-12-02
>
> Dear Reviewer Fejv,
>
> Thank you for your time and thoughtful feedback on our paper. We hope our responses have addressed your concerns, and we kindly request that you consider updating the score accordingly. If there are any remaining issues, please let us know, and we will be happy to provide further clarification.
>
> Thanks!

---

> ### Author Response · Authors · 2024-12-04
>
> Dear Reviewer Fejv,
>
> As the discussion period is ending, we would like to thank you for volunteering your time to review our paper. We hope to have answered all your questions and addressed the rest of the concerns you had.
>
> Thanks!

---

### Author Response · Authors · 2024-12-02
**Paper Update**

We have carefully addressed the reviewers' suggestions and incorporated the following updates into the revised PDF:

- **Reliance on Constraint Verifier**: Added discussion on the wide existence of constraints and their verifiers in lines 462-470.
- **Distribution of Constraint Satisfaction Rate**: Added results in Appendix E and lines 473-476.
- **CommonGen**: Added results in Appendix D.

These updates aim to address the reviewers' feedback comprehensively and strengthen the overall contribution of the paper.

---

### Note · Authors · 2024-12-16

I have read and agree with the venue's withdrawal policy on behalf of myself and my co-authors.